# Comparative Analysis of Image Processing Techniques for Enhanced MRI Image Quality: 3D Reconstruction and Segmentation Using 3D U-Net Architecture

**DOI:** 10.3390/diagnostics13142377

**Published:** 2023-07-14

**Authors:** Chee Chin Lim, Apple Ho Wei Ling, Yen Fook Chong, Mohd Yusoff Mashor, Khalilalrahman Alshantti, Mohd Ezane Aziz

**Affiliations:** 1Faculty of Electronic Engineering & Technology, Universiti Malaysia Perlis, Arau 02600, Perlis, Malaysia; appleling0504@gmail.com (A.H.W.L.); yusoff@unimap.edu.my (M.Y.M.); 2Sport Engineering Research Centre (SERC), Universiti Malaysia Perlis, Arau 02600, Perlis, Malaysia; fook1987@gmail.com; 3Department of Radiology, Universiti Sains Malaysia, Kubang Kerian 16150, Kelantan, Malaysia; khalilalshantti@gmail.com (K.A.); drezane@usm.my (M.E.A.)

**Keywords:** osteosarcoma, bone cancerous cell, deep learning, convolutional neural network, 3D U-Net, MRI, tumor segmentation

## Abstract

Osteosarcoma is a common type of bone tumor, particularly prevalent in children and adolescents between the ages of 5 and 25 who are experiencing growth spurts during puberty. Manual delineation of tumor regions in MRI images can be laborious and time-consuming, and results may be subjective and difficult to replicate. Therefore, a convolutional neural network (CNN) was developed to automatically segment osteosarcoma cancerous cells in three types of MRI images. The study consisted of five main stages. First, 3692 DICOM format MRI images were acquired from 46 patients, including T1-weighted, T2-weighted, and T1-weighted with injection of Gadolinium (T1W + Gd) images. Contrast stretching and median filter were applied to enhance image intensity and remove noise, and the pre-processed images were reconstructed into NIfTI format files for deep learning. The MRI images were then transformed to fit the CNN’s requirements. A 3D U-Net architecture was proposed with optimized parameters to build an automatic segmentation model capable of segmenting osteosarcoma from the MRI images. The 3D U-Net segmentation model achieved excellent results, with mean dice similarity coefficients (DSC) of 83.75%, 85.45%, and 87.62% for T1W, T2W, and T1W + Gd images, respectively. However, the study found that the proposed method had some limitations, including poorly defined borders, missing lesion portions, and other confounding factors. In summary, an automatic segmentation method based on a CNN has been developed to address the challenge of manually segmenting osteosarcoma cancerous cells in MRI images. While the proposed method showed promise, the study revealed limitations that need to be addressed to improve its efficacy.

## 1. Introduction

Osteosarcoma is one of the most prevalent bone tumors, affecting mostly children and adolescents, typically those between 5 and 25 years of age who are experiencing puberty growth spurts [1]. Osteosarcoma usually develops from osteoblasts, which are the cells that make bones, and most commonly affects the distal femur, proximal tibia, and proximal humerus. The American Cancer Society relies on information from the SEER (Surveillance, Epidemiology, and End Results Program) database maintained by the National Cancer Institute (NCI) to provide 5-year relative survival rates statistics for people diagnosed with osteosarcoma between 2010 and 2016 with localized (74%), regional (66%), and distant (27%). If all SEER stages were combined, the 5-year relative survival rate was 60% [2]. Normal, active children and adolescents frequently experience pain and swelling in their limbs, which is more likely to be caused by normal bumps and bruises. Despite the fact that osteosarcoma weakens developing bones, fractures are uncommon, with the exception of the rare telangiectatic osteosarcomas, which weaken bones more than other types of osteosarcoma and are more likely to produce fractures where the osteosarcoma tumor is present [3].

Magnetic resonance imaging (MRI) provides more accurate anatomical information for medical examinations than other medical imaging techniques such as X-rays, ultrasound, and CT images [4]. MRI is an advanced medical imaging technique that provides a wealth of information about the anatomy of human soft tissues. The accuracy of osteosarcoma tumor segmentation in MRI images is important not only for treatment planning for neoadjuvant chemotherapy but also for evaluating subsequent treatment effects. However, the manual method of radiologists outlining malignant tissue from each slice is time-consuming, subjective, and often produces non-repeatable findings. Additionally, the difficulty in identifying osteosarcoma is high due to aspects such as size, location, shape, and texture features. Therefore, this study aims to develop an automatic segmentation algorithm for osteosarcoma tumor cells using convolutional neural networks (CNN) to segment the 3D model for osteosarcoma tumor cells. After the osteosarcoma cancerous cells are segmented automatically, the performance of the auto-segmentation algorithm will be compared to manual segmentation of the region of interest (ROI) of the 3D osteosarcoma tumor cells.

This paper is organized as follows. Section 2 focuses on the literature review of image processing for medical images such as MRI and CT images to review current trends in image processing techniques. Section 3 focuses on the methods used for segmenting the osteosarcoma cancerous cells, which include image pre-processing, image quality assessment (IQA), and segmentation. Section 4 presents the experimental results and discussions of the image pre-processing, IQA, image segmentation, and validation of the proposed model. A conclusion is drawn in Section 5.

## 2. Related Works

This paper presents a deep learning method for segmenting osteosarcoma bone cancerous cells from MRI images. Prior to analysis, the MRI images undergo image processing. Raw images that contain unwanted noise are processed to extract useful information. This image processing involves pre-processing and image segmentation.

### 2.1. Image Pre-Processing

In this paper, a method is introduced to segment osteosarcoma bone cancerous cells using deep learning. Prior to processing the MRI images, image pre-processing is performed to eliminate artifacts and improve signal quality without losing information. Artifacts can significantly affect the MRI image of the osteosarcoma, making it more difficult to analyze and interpret.

Rajeshwari and Sharmila (2013) [5] described a two-phase pre-processing method. In the first phase, a median filter was used to remove film artifacts. In the second phase, an algorithm was introduced to remove unwanted parts using morphological operations. This reduced false positive results in subsequent processing stages and improved results over tracking algorithms that preserve regions of interest and remove film artifacts. This pre-processing method also helps to prevent over-segmentation, which could retain the tumor during further processing.

Suhas and Venugopal (2017) [6] evaluated the performance of MRI image denoising techniques. They applied various filters, including median, Gaussian, max, min, and arithmetic mean filters, to MRI brain and spinal cord images. A new strategy for modifying the existing median filter by adding features is proposed. Experimental results showed that the proposed method, along with the other three image filtering algorithms, improved the root mean square error (RMSE), signal-to-noise ratio (SNR), peak signal-to-noise ratio (PSNR), and other statistical characteristics used to assess output image quality. The proposed method successfully preserved the structural characteristics of the medical image while reducing the majority of the noise.

Intensity normalization is an important preprocessing step for MRI images. In the paper by Mohan and Subashini (2018) [7], six intensity normalization techniques were proposed, including contrast stretching, histogram stretching, histogram equalization, histogram normalization, intensity scaling, and Gaussian kernel normalization. However, according to Loizou et al. (2009) [8], histogram equalization has limited success on medical images due to the removal of minor details. In contrast, Mohan and Subashini (2018) [7] state that histogram equalization is successful, but only when applied to certain types of medical images. To address the limitations of histogram equalization, adaptive and spatially variable processing techniques have been developed. The Wiener filter is an example of such a technique, which can handle spatial and local changes in the image with flexibility.

### 2.2. Segmentation Technique Using CNN in Deep Learning

Deep learning (DL) is a sub-category of machine learning that utilizes multilayer networks to analyze complex patterns in raw image input data. In recent years, DL has proven to be a beneficial tool in various imaging applications.

Singh et al. (2020) [9] reviewed the current applications of interpretable deep learning in different medical imaging tasks. This review discusses various methods, challenges, and areas that require further research in clinical deployment from the practical perspective of deep learning researchers designing systems for clinical end-users.

Debelee et al. (2020) [10] surveyed several deep learning-based techniques for identifying and segmenting tumors in breast cancer, cervical cancer, brain tumors, colon cancer, and lung cancer. Deep learning approaches were found to be the most advanced in tumor identification, segmentation, feature extraction, and classification as a result of the evaluation process.

Çiçek et al. (2016) [11] suggested a volume-to-volume segmentation network called the 3D U-Net, which is a 2D U-Net extension. The 3D U-Net uses dual paths: an analysis path to extract features and a synthesis path for up-sampling to generate full-resolution segmentation. Additionally, the 3D U-Net developed a shortcut link between layers with the same resolution in the early and late stages of the analysis and synthesis path.

Chen et al. (2018) [12] introduced a voxel-wise residual network (VoxResNet), a 3D deep network extension of 2D deep residual learning. VoxResNet provides a skip connection to transfer features from one layer to the next. Although 3D U-Net and VoxResNet provide multiple skip connections to facilitate training, their presence generates a short path from the first layer to the last layer, potentially reducing the network to a relatively simple configuration.

Holbrook et al. (2020) [13] proposed using TensorFlow to segregate soft tissue sarcomas in mice and perform radiomics studies on the obtained MRI datasets using a 3D fully convolutional U-Net network. The cross-entropy loss function was used to achieve the best overall segmentation performance. The dice score for T2-weighted images is 0.861, and the dice score for multi-contrast data is 0.863.

Vaidyanathan et al. (2021) [14] proposed a 3D U-Net to build a deep-learning method for automatic segmentation of the inner ear in MRI by using manually segmented inner ear images as a reference standard. A clinical validation set of eight MRI scans in which the labyrinth’s morphology had changed substantially was also used to validate the model. Across images from three different centers, the 3D U-Net model displayed an accurate mean DSC of 0.8790, a high true positive rate (91.5%), and low false discovery rates and false negative rates (14.8% and 8.49%, respectively).

Feng Liu et al. (2022) [15] proposed OSTransnet, a method for segmenting osteosarcoma MRI images. It combines transformer and U-Net models to address challenges related to fuzzy tumor edge segmentation and overfitting. The technique optimizes the dataset by altering the spatial distribution of noise and applying image rotation. By incorporating channel-based transformers, OSTransnet improves upon the limitations of U-Net and achieves a DSC (dice similarity coefficient) of 0.949, resulting in better segmentation results for osteosarcoma MRI images with blurred tumor borders.

Jia Wu et al. (2022) [16] presented a deep CNN system for osteosarcoma MRI image segmentation. It addresses overfitting due to noisy data and improves generalization. The method involves dataset optimization using Mean Teacher and training with noisy data to enhance robustness. Segmentation utilizes a deep separable U-shaped network (SepUNet) and a conditional random field (CRF). SepUNet effectively segments lesions of different sizes at multiple scales, while CRF refines boundaries. This proposed method was evaluated on 80,000 MRI images from three Chinese hospitals, demonstrating an improved DSC of 0.914.

### 2.3. Summary of Previous Studies

Throughout the research from the previous studies, in the pre-processing process, the contrastivity of MRI images can be enhanced by using histogram equalization or histogram normalization, which are non-linear operations and linear operations, respectively. Other than that, there will be some noise in the MRI images. These noises can be removed by a median filter, which is a non-linear filter, or a Gaussian filter, which is a linear filter. Then, the quality of the processed images can be evaluated by using the peak signal-to-noise ratio (PSNR), mean square error (MSE), and absolute mean brightness error (AMBE). After pre-processing, the segmentation technique by using CNN that was frequently used in previous research is 3D U-Net, which is better at segmenting the 3D model of the region of interest (ROI) as compared to 2D U-Net. Lastly, the accuracy of the automatic segmented ROI can be verified through the dice similarity coefficient (DSC) by comparing it with the manual segmented ROI.

## 3. Results

This section outlines the process flow for carrying out this study. The Python programming language and software used in this study were Google Colaboratory and PyCharm version 2021.2.3 software for simulation using the K80 graphical processing unit with 12 GB of RAM.

Figure 1 shows the overview flow chart that outlines the events in this study. It started acquiring three types of MRI images of the patients who were diagnosed with osteosarcoma. After that, the MRI images were processed with a series of image enhancement methods to improve the quality of the images, so as to increase the accuracy of the segmentation result later on. Then, a convolutional neural network was built by using 3D U-Net architecture to segment the osteosarcoma cancerous cell. Next, the 3D model of the segmented osteosarcoma was visualized to better observe the location and size of the tumor. Lastly, the comparison between manual segmentation and automatic segmentation was carried out using the dice similarity coefficient (DSC) as evaluation criteria.

### 3.1. Image Acquisition

Image acquisition is the initial step in every image processing system, which aims to transform an optical image into an array of numerical data that can be modified on a computer. The information was gathered from the records of the Department of Radiology, Hospital Universiti Sains Malaysia (HUSM), picture archiving and communication system (PACS), and radiology information system using a Philips 3 Tesla Achieva Magnetic Resonance Imaging (MRI) scanner (RIS). The MRI scanner was utilized to collect the MRI images of patients who were diagnosed with osteosarcoma, which are T1W, T2W, and T1W + Gd in the Digital Imaging and Communications in Medicine (DICOM) file type. DICOM is a standard for handling, storing, printing, and transferring medical imaging data [17], and it includes the file format description as well as the network communications protocol. The example MRI images collected are shown in Figure 2.

### 3.2. Image Enhancement

Contrast enhancement techniques are commonly used in medical imaging to improve the visual quality of low-contrast images by emphasizing crucial characteristics or those that are not visible. Contrast enhancement techniques in medical imaging improve the visual quality of low-contrast images by emphasizing important features or invisible details. In this study, contrast-limited adaptive histogram equalization (CLAHE) and contrast stretching (CS) were applied to the MRI images to enhance their contrast, and the best contrast technique was chosen among these two techniques by using the IQA.

CLAHE is a technique used for enhancing local image contrast. CLAHE employed small tiles in the MRI image to compute numerous histograms, each of which compared to a specific area of the image, and then used them to redistribute the image’s brightness or contrast estimation. CLAHE improved the contrast better than normal histogram equalization, which added more detail but amplified noise [18]. A clip limit was then determined for clipping histograms. Each histogram’s height was then redistributed to ensure that it did not exceed the clip limit. The clip limit was calculated as β which can be written in the form as (1) [19], where *M* × *N* is the number of pixels in each region, *L* is the number of grayscales, *α* is a clip factor (0–100), and $s_{max}$ is the maximum allowable slope.
(1)$$\beta =\frac{MN}{L}\left(1+\frac{\alpha }{100}\left(s_{max}-1\right)\right)$$

Contrast stretching is known as normalization, which is a linear operation, meaning the value of the new pixel changes linearly as the value of the original pixel changes [20]. It is a straightforward image enhancement method that involves stretching the range of intensity values to improve image quality. To stretch the image, the higher and lower pixel value limitations over which the image was normalized must be defined, and the existing lowest and highest pixel values should also be identified. As in (2) [21], it was then scaling each pixel in the MRI image, where Pin is the input pixel and Pout is the resulting pixel, a represents the lower limit and b represents the upper limit, and c and d are the current lowest and maximum pixel values, respectively.
(2)$$P_{out}=\left(P_{in}-c\right)\left(\frac{b-a}{d-c}\right)+a$$

### 3.3. Image Denoising

Image quality can be troublesome and poor when acquiring, processing, and storing MRI images. Researchers are still grappling with how to remove noise from original MRI images since noise removal generates artifacts and blurs the images [22]. Different filters can effectively remove different types of noise. In this study, the denoising filters applied include the Gaussian filter and Median filter. The most suitable filter was selected for denoising the MRI images.

The Gaussian filter is a type of linear smoothing filter whose weights are determined by the form of the Gaussian function [23]. The Gaussian filter helps reduce visual noise and minor features drawn from a normal distribution [23]. For image processing, the two-dimensional zero-mean discrete Gaussian function is expressed as (3) [24], where σ is the standard deviation of the distribution, and x and y are the location indices. The value of σ controls the extent of the blurring effect around a pixel by adjusting the variance around the mean value of the Gaussian distribution [24]. It is commonly performed by using a Gaussian kernel to convolve the image. The Gaussian filter is especially useful for filtering images with a lot of noise because the results showed relative independence on the noise features and a significant dependence on the variance value of the Gaussian kernel [25]. The MRI image was denoised using the Gaussian filter by adjusting the standard deviation of the intensity distribution in order to control the blurring effect of the filter, which depends on the effectiveness of the noise to be removed in the MRI image.
(3)$$G_{2D}\left(x,y,\sigma \right)=\frac{1}{2\pi \sigma^{2}}e^{-\frac{x^{2}+y^{2}}{2\sigma^{2}}}$$

The median filter is a sliding window spatial filter that substitutes the median value of all pixel values in the window for the window’s center value [26]. It is a non-linear filtering technique that helps to remove noise and is capable of removing “impulse” noise from either high or low outliers. The standard median filter was given by (4) [26], where *X_i_* and *Y_i_* are the input and output at location *i* of the filter, and *W_i_* is the *r*^th order statistic of the samples inside the window. The MRI image was applied with the median filter to replace the particular pixel with the median value of the sample of its particular window throughout the MRI image.
(4)$$Y_{i}=med\left\{W_{i}\right\}=med\{X_{i}+r:r\in W\}$$

### 3.4. Image Quality Assessment (IQA)

In this study, different techniques were applied for calculating the values of separate parameters such as mean square error (*MSE*), peak signal-to-noise ratio (*PSNR*), and absolute mean brightness error (AMBE).

*MSE* is the most widely used error sensitivity-based image quality assessment, as shown in (5) [23], where M and N are the width and the height of the images, respectively, and *x_ij_* and *y_ij_* are the image gray values of reference image *x* and distorted image *y*. An average of squared intensity differences in every pixel of a reference image and a distorted image was used to calculate it. The lower value of *MSE* indicates there is less error in the image [27].
(5)$$MSE=\frac{1}{MN}\sum_{i=1}^{M}\sum_{j=1}^{N}{\left(x_{ij}-y_{ij}\right)}^{2}$$

The *PSNR* is a quality metric for lossy compressed images. The PSNR is the ratio of the original image’s maximum power to the deformed image’s noisy power. Due to the fact that signals frequently have a wide dynamic range, they are represented in the logarithmic domain. The formula is given as (6), where *MAX* is the maximum possible pixel value of the MRI image [22] and *MSE* is the mean square error. A higher *PSNR* value indicates that the image quality is better.
(6)$$PSNR=10\;\log_{10}\frac{{MAX}^{2}}{MSE}$$

AMBE is an assessment for determining brightness preservation. It assesses the effectiveness of contrast enhancement approaches to maintain the original image’s mean brightness [28]. AMBE uses (7) [28] to calculate the absolute mean brightness difference between the acquired image, *I_in_*, and the pre-processed image, *I_out_*. The lower AMBE value indicates a good performance technique with a high-quality image, and the brightness is better preserved.
(7)$$AMBE=\left|I_{in}-I_{out}\right|$$

### 3.5. Reconstruct MRI Images into 3D Volumes

Before the images passed to the segmentation process, all the DICOM-format images were converted to NIfTI format. The fundamental distinction between DICOM and NIfTI is that NIfTI saves raw image data as a 3D image, whereas DICOM saves raw image data as 2D image slices ([29], p. 4). In addition, NIfTI is modeled as a three-dimensional image, so it is better than DICOM for several deep learning applications as it is easier to manage a single NIfTI file rather than hundreds of DICOM files. For this study, Pycharm software was used for converting DICOM to NIfTI by using the command “dcm2nii” [30]. All input images were rescaled to the same size before training to maintain optimal image features.

### 3.6. Segmentation Model for 3D Volumes

The Python programming language was used to create the 3D U-Net model using the open-source deep learning framework Medical Open Network for AI (MONAI), which was combined with the PyTorch Lightning framework and the PyTorch, Numpy, and Matplotlib libraries. The whole segmentation process in this project is shown in Figure 3.

It started with the transformation of the MRI image to enable the image to be better fitted into the convolutional neural network that will be trained for automatic segmentation later. The MONAI’s “compose” function was used to apply several transforms to the same dataset, which allowed for combining any transformations needed. First, use the “*LoadImaged*” command to load the MRI images and labels from NIfTI format files. Second, the “*AddChanneld*” command added a channel to the MRI image and label. When it came to tumor segmentation, a channel that played the role of background or tumor was required. Third, the “*Orientationd*” command unified the data orientation based on the affine matrix. Fourth, based on the affine matrix, the “*Spacingd*” command was used to adjust the spacing by pixel dimension, “*pixdim*” = (1.5, 1.5, 2.0). This function assisted in changing the voxel dimensions because the dataset of medical images had different voxel dimensions, which were width, height, and depth. Therefore, it is necessary to generalize all of them to the same dimensions. Moreover, the “*CropForegroundd*” command helped to remove all zero borders, allowing the focus to be placed on the valid area of the images and labels. Also, the “*RandCropByPosNegLabeld*” command was included, which helped to randomly crop patch samples from the big image based on the positive-to-negative ratio. The “*RandAffined*” command followed, which efficiently performed rotation, scaling, shearing, translating, and other operations based on the PyTorch affine transform, which was also applied in this pre-processing. Lastly, the command “*EnsureTyped*” converted the Numpy array to PyTorch Tensor, which might be used in subsequent phases.

After the transformation, the data loader was applied to speed up the training process and reduce the memory usage of the graphic processing unit since the MRI datasets used to train were in voxel, which required a longer time in the training process of the neural network. For the purpose of doing so, there were two functions employed, “*CacheDataset*” and “*DataLoader*”, in MONAI dataset managers. Before the first epoch, “*CacheDataset*” performed non-random transforms and prepared cache material in the main process, and then all “*DataLoader*” subprocesses read the same cache content in the main process during training. According to the extent of predicted cache data, preparing cache material may take a long time. In this project, there were two datasets to be built: one to combine the training data with its transforms and the other to combine the validation data with its transforms because there were training and validation sets.

Once the preparation data was performed, the MRI images were ready to train in the 3D U-Net model. A 3D U-Net by MONAI was used in the model design as shown in Figure 4, and the arrows displayed the various operations; the blue boxes showed the feature map at each layer, and the grey boxes described the cropped feature maps from the contracting route. The architecture of the 3D U-Net was roughly divided into two parts, which were the encoder network and decoder network, where each layer had its own encode and decode paths as well as a skip connection between them. Data were down-sampled using strided convolutions in the encoder path, then up-sampled using strided transpose convolutions in the decode path. For the encoder part, it is made up of four blocks, each of which has 3 × 3 × 3 convolutional layers with a PReLU activation function and 2 × 2 × 2 max-pooling layers with strides of two in each dimension. The PReLU activation function used batch normalization to apply a function to the input data in order to boost non-linearity and speed up training. In order to avoid overfitting, the pooling layer downsampled the input values to reduce computing costs and reduce the spatial dimensions of the image. A fully connected layer provided the correlations of the particular class to the high-level features. The number of outputs of the last fully connected layer must be the same as the number of classes [31]. On the other hand, for the decoder path, it was made up of four blocks, each of which contains a 2 × 2 × 2 transposed convolution layer with a stride of 2, followed by two convolutional layers with a size of 3 × 3 × 3, and a PReLU activation function that used batch normalization. The Shortcut connections from equal-resolution layers in the encoder path helped to give the decoder path the necessary high-resolution features. There was also a 1 × 1 × 1 convolution in the final layer with sigmoid output used to reach the feature map with a depth equal to the number of classes, which was 2, where the loss function was determined. Moreover, the high-resolution 3D features in the encoder path were concatenated with up-sampled representations of global low-resolution 3D features in the decoder path to learn and apply local information. The network learned to employ both high-resolution local information and low-resolution global features as a result of this concatenation. During training, the dice loss was used as the loss function, and Adam was used as the optimizer, with a learning rate of 1 × 10^−4^ using backpropagation to find the gradient of the loss function. Table 1 shows an overview of the hyperparameters and their respective values used in this project.

Moreover, the trained model was evaluated quantitatively and qualitatively. The evaluation metrics used to validate qualitatively were DSC. Throughout the training epoch, the model with the highest validation mean DSC for each MRI image type was saved. The average epoch loss was also recorded, which indicated the error occurred in the validation dataset. On the other hand, the validation mean DSC and average epoch loss through the training epoch were used to plot the curve for statistical purposes. In addition, the overlaid image of the MRI image and the predicted output from the 3D U-Net model (Figure 4) were shown to better observe the true or false prediction from the model. Lastly, for better visualization, the 3D volume of the MRI image with the label of the predicted output was displayed.

### 3.7. Image Segmentation Performance Validation

In this study, the suggested evaluation metrics used to validate and compare were dice similarity coefficients (DSC). The DSC, also known as the overlap index [32], was a regularly used performance metric in the field of medical image segmentation. It determined the general similarity rate between a given ground truth label and the expected segmentation output of a segmentation technique. DSC can be expressed as (8) and (9). Where *S_p_* is the predicted segmentation output and *S_g_* is the ground truth label. FP, TP, and FN indicate false positives, true positives, and false negatives, respectively. DSC gave a score between 0 and 1, where 1 denotes the best prediction and indicates that the segmentation result was as expected [32].
(8)$$DSC\left(S_{p},S_{g}\right)=\frac{2TP}{FP+2TP+FN}$$
(9)$$DSC\left(S_{p},S_{g}\right)=\frac{2\left|S_{p}\;\cap \;S_{g}\right|}{\left|S_{p}\right|+\left|S_{g}\right|}$$

## 4. Discussion

### 4.1. Image Pre-Processing

Table 2, Table 3 and Table 4 show contrast-enhanced MRI images by CLAHE and contrast stretching before and after implementing the median filter and Gaussian filter for T1W, T2W, and T1W + Gd, respectively. The goal was to improve the contrast and brightness of the MRI images using these enhancement techniques.

The contrast and brightness of the MRI images had improved for both contrast enhancement methods. For CLAHE, the histogram of intensity value was clipped before computing the cumulative distribution function and distributed uniformly to other bins. As a result, a clear, enhanced MRI image without much noise was obtained. On the other hand, contrast stretching increases the difference between the maximum and minimum intensity values in an image. The remaining intensity values were spread out across the range, making the contrast more noticeable. Therefore, in the resulting image from the contrast stretching, one can clearly observe the contrast between the intensities. From the enhanced MRI images in T1W, T2W, and T1W + Gd, it can be observed that the CLAHE technique can give more detail on the MRI. However, in this study, the region of interest (ROI) was not the details of the MRI images, so these details were considered noise, which caused confusion during the segmentation part due to their similar intensity. Consequently, the contrast stretching technique was more suitable to be used in pre-processing the MRI image in this study.

The MRI images that were processed with the Gaussian filter were more blurry than the MRI images processed with the median filter. This was because the Gaussian filter was a linear type of filter that was more effective in smoothing the image and removing noise. On the other hand, the median filter, which is a non-linear type of filter, showed a better result than the Gaussian filter. This was due to the median filter, which removed thin lines or edges and blurred the image but retained useful details. Moreover, both denoising filters did not show good filtering results in the CLAHE-enhanced MRI images.

### 4.2. Image Quality Assessment

The lower value of MSE indicates there was less error in the image. Overall, from Table 5, the MRI images that were processed with the combination of contrast stretching and median filter scored the lowest MSE value as compared to other combinations. This indicated the combination of contrast stretching and median filter would not affect much on the image quality and would be close to the original MRI image.

The higher the PSNR, the better the image quality after applying filters. In order to calculate the PSNR, we should first have the value of the mean square error (MSE). This was because PSNR was usually expressed in terms of a logarithmic decibel scale to calculate the peak error. From the result in Table 5, as for the PSNR, the combination between the Contrast Stretching and Median filter showed better performance because it got the highest PSNR value.

The lower AMBE indicated better brightness preservation of the image. From Table 5, the AMBE values for the combination of contrast stretching and median filter were very close to zero. So, contrast stretching and median filter showed better brightness preservation than other combinations.

Contrast stretching enhances the contrast between different regions of the cancerous and bone, making its details more visible. By spreading out the intensity values across the range, it increases the separability of different structures and enhances the overall MRI image quality. Thus, this can lead to clearer distinctions between the cancerous region and the surrounding healthy tissue, aiding in their accurate identification and segmentation. This can be crucial for identifying the boundaries, shapes, and other characteristics of bone cancerous cells, which may be critical for accurate segmentation. By applying a Median filter to the MRI images, the filter can effectively remove high-frequency noise components while preserving important details, such as the boundaries and structural characteristics of bone cancerous cells. This preservation of details is essential for accurately delineating and segmenting the cancerous regions. The median filter achieves noise reduction by replacing outlier pixel values, which are likely to be noise, with the median value within a defined neighborhood. This process effectively smoothed the MRI image while retaining the sharpness and integrity of structures, such as the boundaries of the cancerous bone cells. Consequently, the contrast stretching and median filter help improve the visibility and clarity of the cancerous regions within the MRI images.

### 4.3. Reconstruct MRI Images into 3D Model

The enhanced MRI images with contrast stretching and median filter will then be reconstructed into a 3D volume of MRI images, which involves the conversion of DICOM files to NIfTI files. DICOM image slices are stacked to build a 3D representation of the MRI image. The acquired MRI DICOM images had dimensions of length, width, and height, so the resolution of the reconstructed 3D images may vary from the acquired DICOM images. In this study, DICOM images were obtained from the hospital, so each patient may have a different number of slices. The assembled DICOM into a 3D model for T1W, T2W, and T1W + Gd are shown in Table 6. The reconstruction of the 3D model needed DICOM image slices in different planes, such as axial, coronal, and sagittal.

### 4.4. Transformation of 3D Volumes

The transformation techniques applied in this project included image loading, adding channels to the 3D volume and label, data orientation based on the affine matrix, spacing adjustment by pixel dimension, scale intensity ranging in the aspect of contrast, cropping the foreground to remove the zero border, randomly cropping patch samples from the big image, and lastly, the random affine that would be able to perform rotation, scaling, shearing, and translation based on the PyTorch affine transform. These transforms are composed with “Compose” to create a fast pipeline. Based on the result obtained as shown in Table 7, the images were resampled to a voxel size of 1.5, 1.5, and 2.0 mm in each dimension to avoid any dimension error that might occur when loading the images into the network. In addition, the 3D sub-volumes were also padded to sizes of 96, 96, and 96 to make sure the input sizes of the images to be loaded into the neural network were consistent. Moreover, the cropping of the foreground and random cropping of patch samples were also effective, as shown in Table 7, where the ROI of the images was amplified to better focus on the ROI when training the neural network. This transformation method needed to be applied to both the input image and the segmentation mask.

### 4.5. Quantitative and Qualitative Evaluation of 3D U-Net Model

The 3D U-Net was trained and validated with T1W, T2W, and T1W + Gd MRI image datasets through the implementation of Adam as the optimizer algorithm and Sigmoid as the activation function on the GPU in Google Colaboratory. The 46 MRI images for each MRI image type were split into 36 MRI images for training, 5 MRI images for validation, and 5 MRI images for testing. These three types of MRI datasets took about 6 hours to train the 3D U-Net model, as shown in Table 8.

Based on Table 8, the proposed algorithm obtained a good DSC on all the MRI image types. The T2W and T1W + Gd achieved excellent segmentation results with approximately 85% and 87% of the validation mean DSC at epochs 792 and 700, respectively, while the T1W obtained a good result with around 83% of the validation mean DSC at epoch 786. The validation mean DSC of the T1W image dataset was the lowest among these three datasets. This is because T2W optimally shows fluid and abnormalities such as tumors, inflammation, and trauma [33], and T1W + Gd inhibits the fat signal in T1W, which then increases its significance in assessing tumor vascularization [34]. In contrast, T1W optimally shows normal soft tissue anatomy and fat [33]. For this reason, T1W might have a low contrast for the tumors to be identified as compared to T2W and T1W + Gd. The dice loss function is defined as 1 minus the DSC in order to indicate the loss function’s convergence. The epoch average dice loss produced by these three MRI datasets was approximately between 0.15 and 0.17, as shown in Table 8. The smaller the value of the epoch average dice loss, the less error there is in the validation dataset.

Different examples of different MRI image types after implementing the proposed segmentation model are shown in Table 9. Based on the segmentation result, the output test image for each T1W, T2W, and T1W + Gd could be compared with the ground truth. There were 3 samples with their respective 80th slice of predicted segmentation output from the proposed segmentation method’s qualitative results. The output of samples 1 and 2 showed that the majority of the tumor was accurately segmented for T2W and T1W + Gd, with slight border and small hole errors. However, it can be clearly observed that the output of T1W had a false negative as compared to the ground truth, in which some areas of the ROI were not predicted. Other than that, there was another common error shown in sample 3, which was a false positive. It can be observed that there should be no tumor being segmented in sample 3. Yet, a small false-positive tumor was found in the output of the T2 image due to the presence of tissue with a similar look.

In order to have a better observation of the result obtained, the overlaid slices of sample 3 from slices 60th, 70th, 80th, 90th, and 100th were used as representative examples for each T1W, T2W, and T1W + Gd, as shown in Table 10, Table 11 and Table 12. The last column of the table shows the overlaid slices between the ground truth and the predicted output, where the red tint denotes the output predicted by the model, the white tint denotes the true ground truth, and the black color indicates the background.

The predicted output of the 60th slice of each type of MRI image did not show any tumor, which was segmented correctly as compared with ground truth for each T1W, T2W, and T1W + Gd as in Table 12. From the overlaid slice of the 70th slice for these three types of MRI images, it was obviously noticed that there was a bit of false-negative tumor wrongly predicted by the trained model. However, in the 70th slice of these three types of MRI images, T2W performed the best among these MRI image types because it predicted the location of a small detail correctly, while the other two image types did not. In addition, there was also a small false-positive tumor shown in these overlaid slices. Moreover, from the observed results of 80th, 90th, and 100th, even though some of the slices of ground truth were not fully overlapped with the output, most of the slices as shown were predicted almost the same, with a small false-negative tumor as compared with ground truth.

### 4.6. Visualisation of the Predicted Output

In order to have a clearer visualization of the segmented data, the TensorBoard 3D plugin was used to view the entirety of their three-dimensional deep-learning model output. Thereby, the size and position of the tumor can be clearly observed through this 3D viewer. Figure 5 shows the 3D model output, where the red tint denotes the predicted tumor from the trained model.

### 4.7. Comparision DSC with Previous Research Works

Based on the comparison of DSC with other researchers’ segmentation models for osteosarcoma MRI images in Figure 6, OSTransnet (by F. Liu, 2022) [15] stands out for its superior segmentation performance, incorporating contextual information and edge enhancement, although its limitations were not outlined. PESNet (Baolong Lv, 2022) enhances tumor localization and segmentation accuracy through a priori generation and feature enrichment networks, but it has a slightly higher computation time than U-Net [35]. MSFCN allows for multi-scale feature integration and can handle objects at various scales. It can be useful for tasks like semantic segmentation and object detection. However, MSFCN may not be explicitly designed for 3D or volumetric data. Its performance can vary depending on the specific task and dataset. SepUNet (by J. Wu, 2022) [16] is a powerful model for osteosarcoma segmentation, providing improved accuracy and efficiency. It excels at handling tumors of various sizes while maintaining a small parameter count, making it accessible and computationally efficient. However, specific limitations were not mentioned. SLIC-S (by E. B. Kayal, 2020) [36] provides efficient superpixel-based segmentation into compact, spatially connected regions. But it may struggle with fine-grained details and be sensitive to the parameter settings of the superpixel generation process. Our 3D U-Net is designed for volumetric segmentation, utilizing 3D convolutions and skip connections to capture spatial information and fuse multi-scale features. However, it requires high memory and computational resources due to volumetric data processing, which provided a slightly lower 0.8762 of DSC. FCM (by E. B. Kayal, 2020) [35] offers flexibility in unsupervised clustering and deals effectively with complex data distributions and partial volume effects. But it may face computational complexity and sensitivity to parameters such as ambiguous data, which caused the lowest DSC of 0.87. From the comparison, SepUNet, PESNet, FCM, and SLIC-S are primarily used for 2D segmentation, whereas 3D U-Net and OSTransnet are tailored for 3D segmentation tasks. They are specifically designed to process 3D volumes or stacks of medical images, allowing for the segmentation of objects or regions in the volumetric space. These models leverage the spatial information and contextual cues present in three-dimensional data to achieve accurate and comprehensive segmentations.

## 5. Conclusions

The proposed segmentation model was 3D U-Net using MONAI. After a series of training epochs, the model with the highest dice similarity coefficient value was saved for these three different types of MRI images. In the perspective of quantitative analysis, the T1W, T2W, and T1W + Gd achieved good validation mean DSC of 83.75%, 85.45%, and 87.62%, respectively, at epochs 786, 792, and 700. In addition, the epoch average dice loss decreased as the training went over the epoch and got closer to zero, which was about 0.15 to 0.17. From the perspective of qualitative analysis, the overlaid images of the predicted output and ground truth showed high overlapping, with a small false negative and false positive appearing. Although the relative results were good, some of the qualitative data indicate that the proposed method was still constrained by ill-defined borders, missing lesion portions, and other confounding factors. Furthermore, even though all training photos have the determined minimum lesion size maximized, it is possible that some small lesions are still filtered out during testing. In summary, the larger tumor segmentation performed significantly better with the model trained for these three datasets.

## Figures and Tables

**Figure 1 diagnostics-13-02377-f001:**
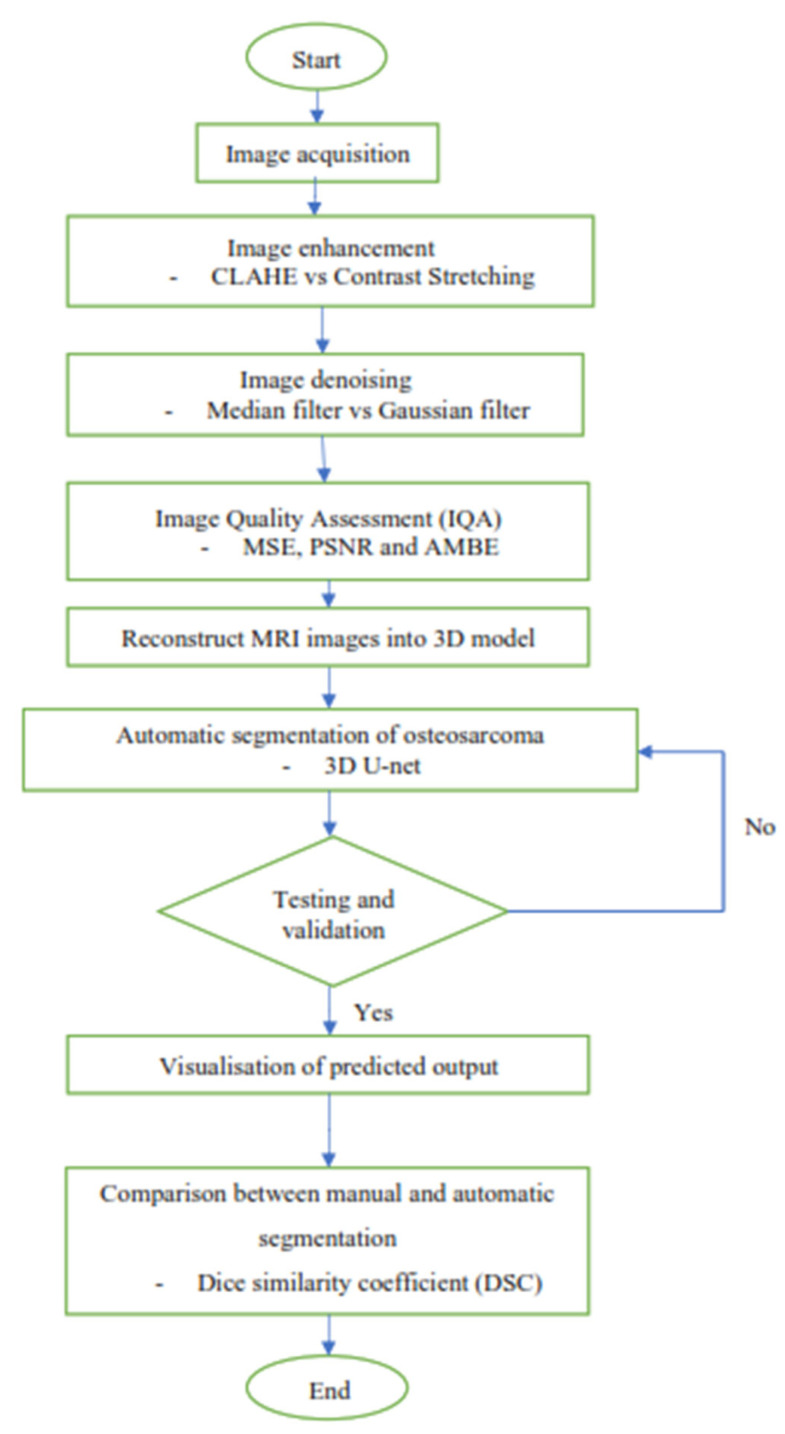
Overview flow chart.

**Figure 2 diagnostics-13-02377-f002:**
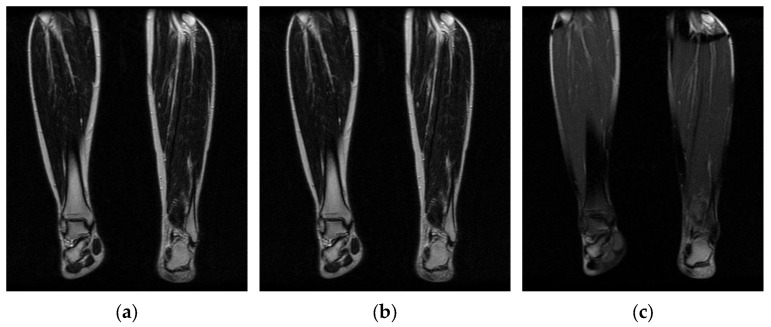
Types of MRI images. (**a**) T1-weighted MRI image (T1W); (**b**) T2-weighted MRI image (T2W); (**c**) T1-weighted MRI image with an injection of gadolinium (T1W + Gd).

**Figure 3 diagnostics-13-02377-f003:**
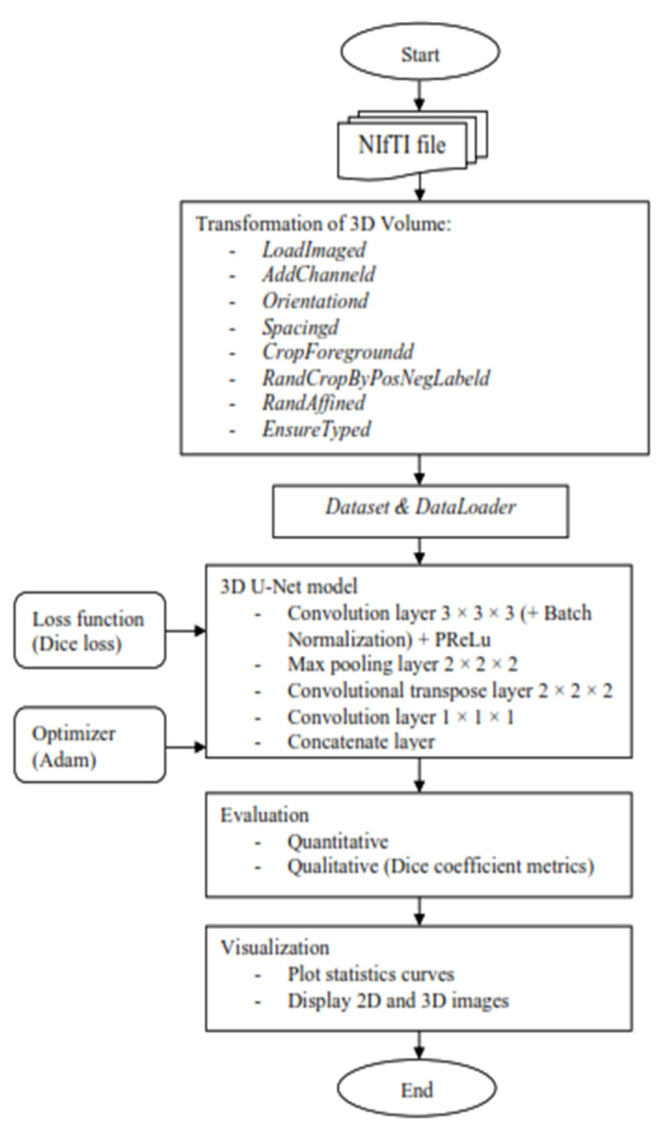
Flow chart of the segmentation model for 3D volume.

**Figure 4 diagnostics-13-02377-f004:**
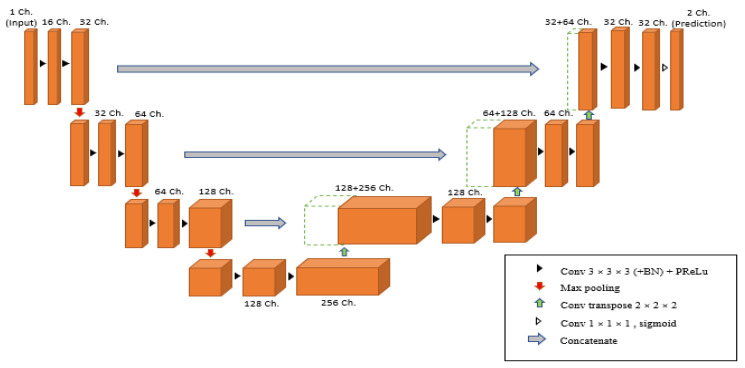
Three-dimensional (3D) U-Net architecture.

**Figure 5 diagnostics-13-02377-f005:**
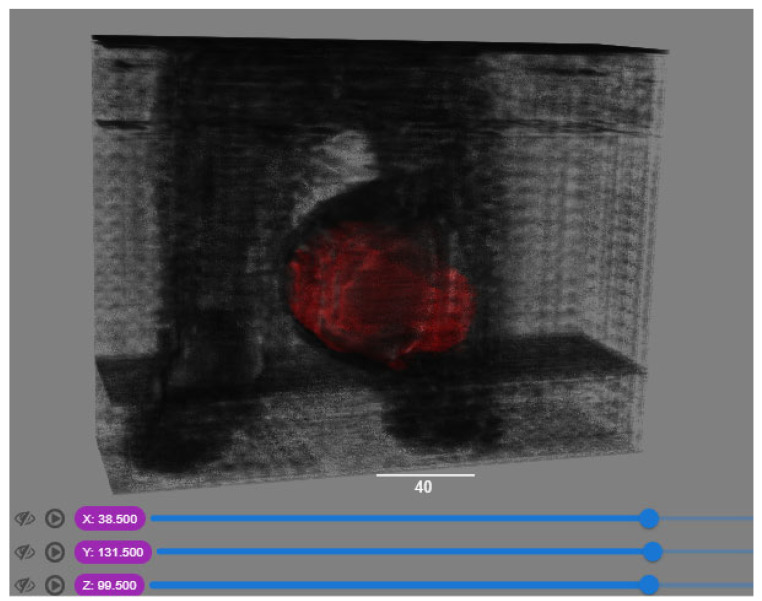
Three-dimensional model output (red tint denotes the predicted tumor from the trained 3D U-net Model).

**Figure 6 diagnostics-13-02377-f006:**
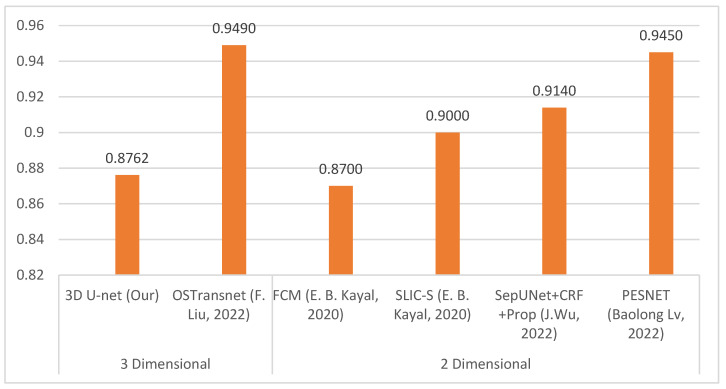
Comparative of DSC between other researchers’ segmentation models for osteosarcoma MRI images [15,16,35,36].

**Table 1 diagnostics-13-02377-t001:** Overview of the hyperparameters set for training.

Hyperparameters	Values
Data split ratio	8:1:1
Maximum epochs	800
Batch size	2
Optimizer	Adam
Loss function	Dice loss
Activation function	PReLU
Total number of parameters	4,808,917

**Table 2 diagnostics-13-02377-t002:** Contrast-enhanced T1W MRI image after implementing Gaussian filter and median filter.

T1W MRI Image after Contrast Enhancement	Combination of Pre-Processing Techniques
CLAHE + Gaussian Filter	CLAHE + Median Filter	Contrast Stretching + Gaussian Filter	Contrast Stretching + Median Filter
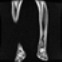	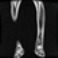	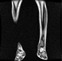	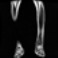	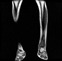
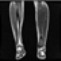	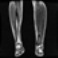	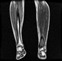	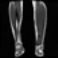	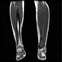

**Table 3 diagnostics-13-02377-t003:** Contrast-enhanced T2W MRI image after implementing Gaussian filter and median filter.

T2W MRI Image after Contrast Enhancement	After Pre-Processing of T2W
CLAHE + Gaussian Filter	CLAHE + Median Filter	Contrast Stretching + Gaussian Filter	Contrast Stretching + Median Filter
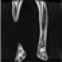	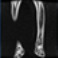	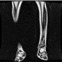	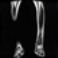	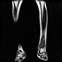
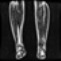	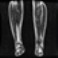	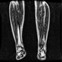	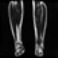	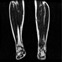

**Table 4 diagnostics-13-02377-t004:** Contrast-enhanced T1W + Gd MRI image after implementing Gaussian filter and median filter.

T1W + Gd MRI Image after Contrast Enhancement	After Pre-Processing of T1W + Gd
CLAHE + Gaussian Filter	CLAHE + Median Filter	Contrast Stretching + Gaussian Filter	Contrast Stretching + Median Filter
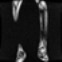	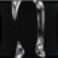	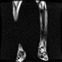	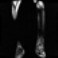	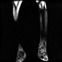
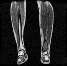	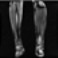	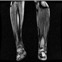	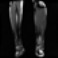	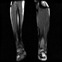

**Table 5 diagnostics-13-02377-t005:** Comparison of MSE, PSNR, and AMBE between T1W, T1W + Gd and T2W implemented with CLAHE or contrast stretching and Gaussian filter or median filter.

		MSE			PSNR			AMBE		
		T1W	T1W + Gd	T2W	T1W	T1W + Gd	T2W	T1W	T1W + Gd	T2W
CLAHE + Gaussian filter	MRI 1	99.5270	98.0830	110.4348	28.1514	28.2149	27.6997	0.06093	0.07008	0.08086
	MRI 2	77.6347	73.5121	106.9793	29.2302	29.4672	27.8378	0.07691	0.08538	0.10174
	MRI 3	88.3859	86.8852	108.9995	28.6670	28.7413	27.7566	0.04468	0.06080	0.07764
	MRI 4	93.6140	105.3020	108.0355	28.4174	27.9064	27.7951	0.04200	0.05408	0.05596
	MRI 5	94.4030	100.5126	110.5212	28.3809	28.1086	27.6964	0.04347	0.05096	0.05848
	AVG	90.7129	92.8589	108.9940	28.5693	28.4876	27.7571	0.05359	0.06426	0.07493
CLAHE + Median filter	MRI 1	88.8244	88.3116	99.6605	28.6455	28.6706	28.1456	0.05805	0.06831	0.07490
	MRI 2	77.0120	72.0549	100.0506	29.2652	29.5542	28.1286	0.07475	0.08146	0.09748
	MRI 3	81.6804	82.0138	97.0345	29.0096	28.9919	28.2615	0.04361	0.05940	0.06829
	MRI 4	76.8616	91.1990	100.1020	29.2737	28.5309	28.1264	0.38308	0.05018	0.05279
	MRI 5	83.4949	87.9688	101.8403	28.9142	28.6875	28.0516	0.04084	0.04781	0.05486
	AVG	81.5747	84.3096	99.7376	29.0216	28.8870	28.1427	0.1201	0.0614	0.0697
Contrast Stretching + Gaussian filter	MRI 1	22.5087	18.8500	24.1599	34.6073	35.3777	34.1599	0.01251	0.01570	0.02479
	MRI 2	40.4818	38.7746	38.2741	32.0582	32.2453	32.3018	0.04688	0.04477	0.04038
	MRI 3	22.0822	21.9366	20.2009	34.6904	34.7191	35.0771	0.01152	0.01697	0.01564
	MRI 4	7.3680	55.7069	8.9038	39.4573	30.6717	38.6351	0.01216	0.03440	0.01196
	MRI 5	7.4555	15.0778	9.9041	39.4060	36.3474	38.1726	0.01061	0.02000	0.01385
	AVG	19.9792	30.0692	20.2886	36.0438	33.8722	35.6693	0.01874	0.02647	0.02132
Contrast Stretching + Median filter	MRI 1	21.7134	18.5888	23.6220	34.7635	35.4383	34.3976	0.01197	0.01522	0.02406
	MRI 2	39.5238	38.6358	36.7550	32.1622	32.2609	32.4776	0.04632	0.04444	0.03961
	MRI 3	20.9480	20.8487	19.3480	34.9194	34.9400	35.2644	0.01099	0.01660	0.01492
	MRI 4	7.6674	52.1598	9.1981	39.2844	30.9575	38.4938	0.01136	0.03300	0.01047
	MRI 5	7.5921	14.8690	10.3506	39.3272	36.4080	37.9812	0.00986	0.01906	0.01293
	AVG	19.4889	29.0204	19.8547	36.0913	34.0009	35.7229	0.01810	0.02566	0.02040

Note: The best reading of each categories is highlighted and compare between MRI images.

**Table 6 diagnostics-13-02377-t006:** Assembling DICOM into 3D model.

Type of MRI	Plane	DICOM Image	3D Model in NIfTI
T1W	Axial	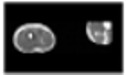	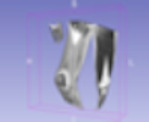
Coronal	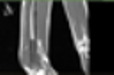
Sagittal	
T1W + Gd	Axial	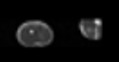	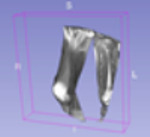
Coronal	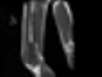
Sagittal	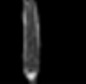
T2W	Axial	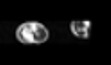	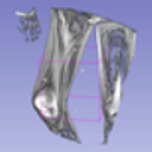
Coronal	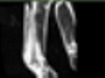
Sagittal	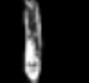

**Table 7 diagnostics-13-02377-t007:** Transformation of 3D volumes.

Slice of Original Image	Transformation for Ground Truth	After Transformation		
		T1W	T1W + Gd	T2W
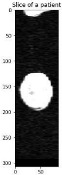	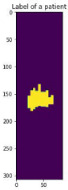	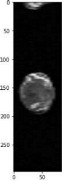	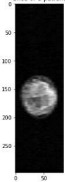	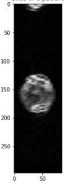
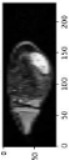	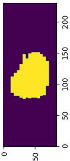	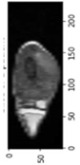	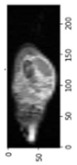	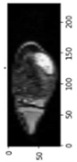
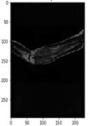	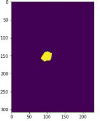	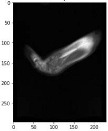	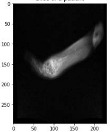	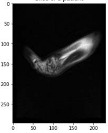

Note: Yellow indicates the ROI of MRI images.

**Table 8 diagnostics-13-02377-t008:** The training time, mean dice similarity coefficient (DSC) and the epoch average dice loss of model of different MRI image types.

Types of MRI Image	Epoch	Training Time (Second)	Mean DSC	Epoch Average Dice Loss
T1W	786	20,194.563	0.8375	0.1709
T2W	792	20,429.427	0.8545	0.1563
T1W + Gd	700	20,020.069	0.8762	0.1534

**Table 9 diagnostics-13-02377-t009:** Segmentation output (highlighted in yellow) of the osteosarcoma cancerous cell.

Sample No. (MRI Slice)	Input	Ground Truth	Output		
			T1W	T2W	T1W + Gd
1 (80th slice)	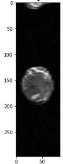	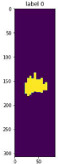	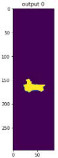	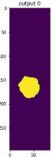	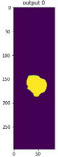
2 (80th slice)	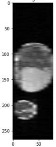	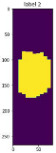	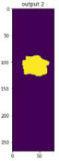	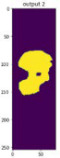	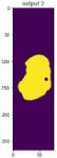
3 (80th slice)	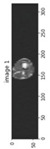	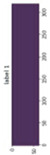	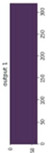	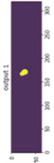	

**Table 10 diagnostics-13-02377-t010:** Overlaid slices for output of the osteosarcoma cancerous cell in T1W MRI image for sample 3.

MRI Slice	Input	Ground Truth (G)	Output (O)	Overlaid Slices
Ground Truth	Output	Ground Truth and Output, (G ∩ O)
**60th**	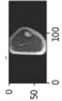	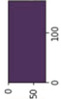	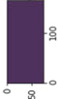	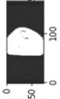	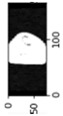	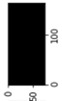
**70th**	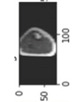	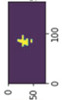	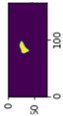	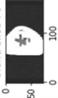	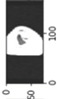	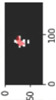
**80th**	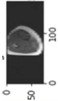	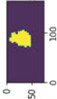	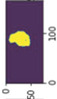	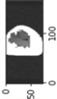	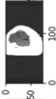	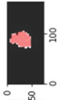
**90th**	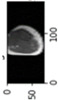	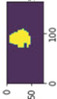	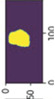	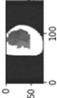	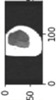	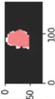
**100th**	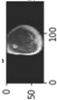	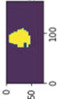	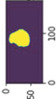	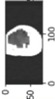	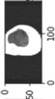	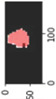

Note: Ground truth (G): yellow denotes manually segmented cancerous cell by medical doctors; Output (O): yellow denotes segmented cancerous cell output predicted by 3D U-net model; Ground Truth and Output under overlaid slide (**G** ∩ **O**): red denotes overlapping area of the ground truth and output, white denotes non-overlapping area of the ground truth and output, black denotes the background.

**Table 11 diagnostics-13-02377-t011:** Overlaid slices for output of the osteosarcoma cancerous cell in T1W + Gd MRI image for sample 3.

MRI Slice	Input	Ground Truth	Output	Overlaid Slices
Ground Truth	Output	Ground Truth and Output
**60th**	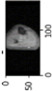	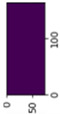	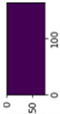	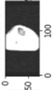	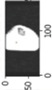	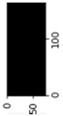
**70th**	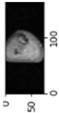	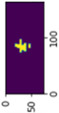	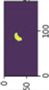	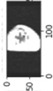	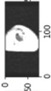	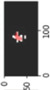
**80th**	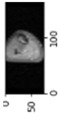	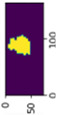	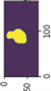	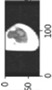	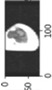	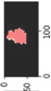
**90th**	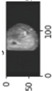	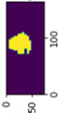	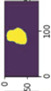	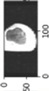	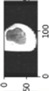	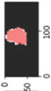
**100th**	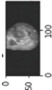	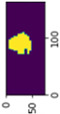	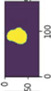	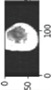	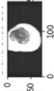	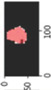

Note: Ground truth (G): yellow denotes manually segmented cancerous cell by medical doctors; Output (O): yellow denotes segmented cancerous cell output predicted by 3D U-net model; Ground Truth and Output under overlaid slide (**G** ∩ **O**): red denotes overlapping area of the ground truth and output, white denotes non-overlapping area of the ground truth and output, black denotes the background.

**Table 12 diagnostics-13-02377-t012:** Overlaid slices for output of the osteosarcoma cancerous cell in T2W MRI image sample 3.

MRI Slice	Input	Ground Truth	Output	Overlaid Slices
Ground Truth	Output	Ground Truth and Output
60th	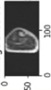	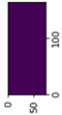	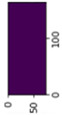	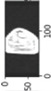	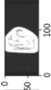	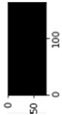
70th	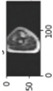	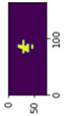	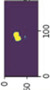	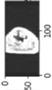	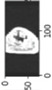	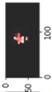
80th	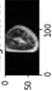	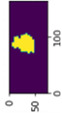	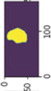	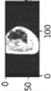	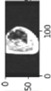	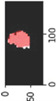
90th	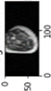	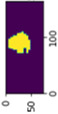	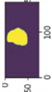	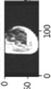	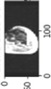	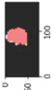
100th	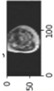	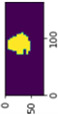	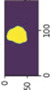	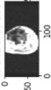	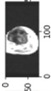	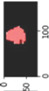

Note: Ground truth (G): yellow denotes manually segmented cancerous cell by medical doctors; Output (O): yellow denotes segmented cancerous cell output predicted by 3D U-net model; Ground Truth and Output under overlaid slide (**G** ∩ **O**): red denotes overlapping area of the ground truth and output, white denotes non-overlapping area of the ground truth and output, black denotes the background.

## Data Availability

Data are unavailable due to privacy or ethical restrictions; a statement is still required.

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
