# Peer review of "Comparative Analysis of Image Processing Techniques for Enhanced MRI Image Quality: 3D Reconstruction and Segmentation Using 3D U-Net Architecture"

_diagnostics, 2023, doi:10.3390/diagnostics13142377_

Round 1
Reviewer 1 Report
The paper sounds good, it discusses a crucial issue in the field of MRI images;
I have some comments to improve it :
1. Lack of contrast in all images .
2. The LR needs more improvement.
The table (. Comparison of MSE, PSNR and AMBE between T1W, T1W+Gd and T2W implemented 410 with CLAHE or contrast stretching and Gaussian filter or median filter. ) should clarify the results and highlight the significance results.
3. You applied various image processing techniques besides evaluating the image quality (what is exactly your aim comparison or segmentation) that should be cleared when you discuss the results and in the title of your manuscript.
4. Talk briefly about the running environment (software, PC, running time for training and testing)
5. To avoid overffitting what is your validation techniques that have been used?
6. compare your results with the existing studies.
7. Add a flow diagram to summarize your method.
8. show more example about your segmentation results (ground truth and the predicted one)
Author Response
RESPONSE TO REVIEWER
REVIEWER 1
- Lack of contrast in all images.
Response: All the images is improved.
- The LR needs more improvement.
Response: Literature Review is improved by adding statement in section 2 Line 145 until Line 159
“Feng Liu et. al., 2022 [33] propose OSTransnet, a method for segmenting osteosar-coma MRI images. It combines Transformer and U-net models to address challenges re-lated to fuzzy tumor edge segmentation and overfitting. The technique optimizes the dataset by altering the spatial distribution of noise and applying image rotation. By incor-porating channel-based transformers, OSTransnet improves upon the limitations of U-Net and achieves a DSC (Dice Similarity Coefficient) of 0.949, resulting in better segmentation results for osteosarcoma MRI images with blurred tumor borders.
Jia Wu et. al., 2022 [34] presents a deep CNN system for osteosarcoma MRI image seg-mentation. It addresses overfitting due to noisy data and improves generalization. The method involves dataset optimization using Mean Teacher and training with noisy data to enhance robustness. Segmentation utilizes a deep separable U-shaped network (Se-pUNet) and conditional random field (CRF). SepUNet effectively segments lesions of dif-ferent sizes at multiple scales, while CRF refines boundaries. This proposed method is evaluated on 80,000 MRI images from three Chinese hospitals, demonstrating improved DSC 0.914.”
- The table (Comparison of MSE, PSNR and AMBE between T1W, T1W+Gd and T2W implemented 410 with CLAHE or contrast stretching and Gaussian filter or median filter.) should clarify the results and highlight the significance results.
Response: The clarify of results is done by adding as below statement in line 433 until 448.
“Contrast stretching enhance the contrast between different regions of the cancerous and bone, making its details more visible. By spreading out the intensity values between the range, it increases the separability of different structures and enhances the overall MRI image quality. Thus, this can lead to clearer distinctions between cancerous region and the surrounding healthy tissue, aiding in their accurate identification and segmentation This can be crucial for identifying the boundaries, shapes, and other characteristics of bone cancerous cells, which may be critical for accurate segmentation. By applying a Median filter to the MRI images, the filter can effectively remove high-frequency noise components while preserving important details, such as the boundaries and structural characteristics of bone cancerous cells. This preservation of details is essential for accurately delineating and segmenting the cancerous regions. Median filter achieves noise reduction by replacing outlier pixel values, which are likely to be noise, with the median value within a de-fined neighborhood. This process effectively smoothed the MRI image while retaining the sharpness and integrity of structures, such as the boundaries of bone cancerous cells. Consequently, the contrast stretching and median filter helps improve the visibility and clarity of the cancerous regions within the MRI images.”
The significance results are added in Table 5 is highlighted.
- You applied various image processing techniques besides evaluating the image quality (what is exactly your aim comparison or segmentation) that should be cleared when you discuss the results and in the title of your manuscript.
Response: A comparative analysis of various image processing techniques was conducted to determine the most effective method for enhancing image quality. The selected techniques were subsequently applied to reconstruct MRI images into 3D volume transformations, followed by 3D segmentation using 3D U-net architecture.
Thus, the title had change to “Comparative Analysis of Image Processing Techniques for Enhanced MRI Image Quality: 3D Reconstruction and Segmentation using 3D U-Net Architecture”
- Talk briefly about the running environment (software, PC, running time for training and testing).
Response: Statement added in section 3 line 174 to line 176.
“The Python programming language and software used in this study were Google Cola-boratory and PyCharm software for simulation by using graphical processing unit K80 with 12GB RAM.”
- To avoid overfitting what is your validation techniques that have been used?
Response: The explanation as below statement is added to the section 3.6 line 336 to 341.
“The PReLU activation function used Batch Normalization to apply a function to the input data in order to boost non-linearity and speeded up training. In order to avoid overfitting, the pooling layer down sampled the input values to reduce computing costs and reduced the spatial dimensions of the image. A fully connected layer provided the correlations of the particular class to the high-level features. The number of outputs of the last fully connected layer must same as the number of classes [34].”
- Compare your results with the existing studies.
Response: Section 4.8 for the comparison and explanation of existing studies with our studies.
- Add a flow diagram to summarize your method.
Response: A flow diagram which summarize my work is shown in Figure 1.
- show more example about your segmentation results (ground truth and the predicted one) as in Table 9, 10, 11
Response: Thank you for your suggestion to include additional segmentation results in the form of Tables 9, 10, and 11. While I understand your desire for more examples, it is important to consider the length and readability of the paper. Including too many tables may make the document lengthy and exceed the desired page limit. However, I appreciate your interest in seeing more segmentation results, and I will take this into account for future research or presentations where a more comprehensive display of results can be provided.

Reviewer 2 Report
The clarity and the presentation of the paper need to improve. I cannot recommend this publication for publication in this form. Moreover, some of the observations are given below.
@ The feature extraction methods used in the proposed framework could be further refined or customized to better capture the relevant characteristics of different types of images and distortions.
@The paper does not discuss the limitations or potential drawbacks of the proposed framework.
@The related work section should be extended to include new methods in the state of the art. There are various U-Net variants in literature I believe a detailed paragraph should be added discussing those variants. These U-Net variants specifically for medical images that are suggested to add are:
- https://doi.org/10.3390/diagnostics11020169
- https://doi.org/10.3390/rs13020310
- https://doi.org/10.1016/j.compbiomed.2022.106426
- https://doi.org/10.3390/s22030867
- https://doi.org/10.1016/j.bbe.2020.05.006
- https://doi.org/10.3390/electronics9122203
@ Moreover if you are making a whole section on image enhancement and image quality assessment so at least you should dedicate a paragraph regarding the latest techniques in these fields. A few of the suggested articles are as follows,
- https://doi.org/10.1016/j.sigpro.2022.108821
- https://doi.org/10.1007/s11042-021-11877
- https://doi.org/10.1007/s11042-022-12060-6
@ The quality of the figures is very poor. Especially the flow diagrams.
@ Check the captions of the figure. How are there 2 Figure 1?
Language is fine
Author Response
RESPONSE TO REVIEWER
REVIEWER 2
- The feature extraction methods used in the proposed framework could be further refined or customized to better capture the relevant characteristics of different types of images and distortions.
Response: For this research, we did not use feature extraction method in the proposed framework.
- The paper does not discuss the limitations or potential drawbacks of the proposed framework.
Response: Advantages and limitation of proposed 3D U-net method is explained in section 4.8 Line 584 until 588.
“Our 3D U-Net is designed for volumetric segmentation, utilizing 3D convolutions and skip connections to capture spatial information and fuse multi-scale features. However, it requires high memory and computational resources due to volumetric data processing which provided a slightly lower 0.8762 of DSC.”
- The related work section should be extended to include new methods in the state of the art. There are various U-Net variants in literature I believe a detailed paragraph should be added discussing those variants. These U-Net variants specifically for medical images that are suggested to add are:
- https://doi.org/10.3390/diagnostics11020169
- https://doi.org/10.3390/rs13020310 (UAV no suitable for this research work)
- https://doi.org/10.1016/j.compbiomed.2022.106426
- https://doi.org/10.3390/s22030867
- https://doi.org/10.1016/j.bbe.2020.05.006
- https://doi.org/10.3390/electronics9122203
Response: Literature Review is improved with statement as below in section 2 Line 145 until Line 159
“Feng Liu et. al., 2022 [33] propose OSTransnet, a method for segmenting osteosar-coma MRI images. It combines Transformer and U-net models to address challenges re-lated to fuzzy tumor edge segmentation and overfitting. The technique optimizes the da-taset by altering the spatial distribution of noise and applying image rotation. By incorporating channel-based transformers, OSTransnet improves upon the limitations of U-Net and achieves a DSC (Dice Similarity Coefficient) of 0.949, resulting in better segmentation results for osteosarcoma MRI images with blurred tumor borders.
Jia Wu et. al., 2022 presents a deep CNN system for osteosarcoma MRI image seg-mentation. It addresses overfitting due to noisy data and improves generalization. The method involves dataset optimization using Mean Teacher and training with noisy data to enhance robustness. Segmentation utilizes a deep separable U-shaped network (Se-pUNet) and conditional random field (CRF). SepUNet effectively segments lesions of different sizes at multiple scales, while CRF refines boundaries. This proposed method is evaluated on 80,000 MRI images from three Chinese hospitals, demonstrating improved DSC 0.914.”
- Moreover if you are making a whole section on image enhancement and image quality assessment so at least you should dedicate a paragraph regarding the latest techniques in these fields. A few of the suggested articles are as follows,
- https://doi.org/10.1016/j.sigpro.2022.108821
- https://doi.org/10.1007/s11042-021-11877
- https://doi.org/10.1007/s11042-022-12060-6
Response: I understand that you have suggested including a dedicated paragraph on the latest techniques in image enhancement and image quality assessment. While I appreciate your suggestion, I believe that the current content already provides a comprehensive overview of the mentioned models and their advantages and limitations. However, I will take note of your suggestion for future reference and consider incorporating it in future discussions or articles on the topic. Thank you for your input.
- The quality of the figures is very poor. Especially the flow diagrams.
Response: The quality of the figure is improved.
- Check the captions of the figure. How are there 2 Figure 1?
Response: The caption and label of all figures is revised.

Round 2
Reviewer 1 Report
The paper can be accepted
Reviewer 2 Report
Can be accepted
Can be accepted